# DNA Double-Strand Break-Related Competitive Endogenous RNA Network of Noncoding RNA in Bovine Cumulus Cells

**DOI:** 10.3390/genes14020290

**Published:** 2023-01-22

**Authors:** Jian-Bo Liu, Jia-Bao Zhang, Xiang-Min Yan, Peng-Gui Xie, Yao Fu, Xu-Huang Fu, Xu-Lei Sun, Dong-Xu Han, Sheng-Peng Li, Yi Zheng, Yan Gao, Nam-Hyung Kim, Bao Yuan, Hao Jiang

**Affiliations:** 1Department of Laboratory Animals, College of Animal Sciences, Jilin University, Changchun 130062, China; 2Experimental Testing Center, Jilin Agricultural Science and Technology University, Jilin 132101, China; 3Institute of Animal Husbandry, Xinjiang Academy of Animal Husbandry, Urumqi 830057, China; 4Yili Vocational and Technical College, Yili 835000, China; 5Department of Animal Science, Chungbuk National University, Cheongju 361-763, Republic of Korea

**Keywords:** DNA double-strand breaks, DDR, ncRNA, ceRNA, cumulus cells, bovine

## Abstract

(1) Background: DNA double strand breaks (DSBs) are the most serious form of DNA damage that affects oocyte maturation and the physiological state of follicles and ovaries. Non-coding RNAs (ncRNAs) play a crucial role in DNA damage and repair. This study aims to analyze and establish the network of ncRNAs when DSB occurs and provide new ideas for next research on the mechanism of cumulus DSB. (2) Methods: Bovine cumulus cells (CCs) were treated with bleomycin (BLM) to construct a DSB model. We detected the changes of the cell cycle, cell viability, and apoptosis to determine the effect of DSBs on cell biology, and further evaluated the relationship between the transcriptome and competitive endogenous RNA (ceRNA) network and DSBs. (3) Results: BLM increased γH2AX positivity in CCs, disrupted the G1/S phase, and decreased cell viability. Totals of 848 mRNAs, 75 long noncoding RNAs (lncRNAs), 68 circular RNAs (circRNAs), and 71 microRNAs (miRNAs) in 78 groups of lncRNA–miRNA–mRNA regulatory networks, 275 groups of circRNA–miRNA–mRNA regulatory networks, and five groups of lncRNA/circRNA–miRNA–mRNA co-expression regulatory networks were related to DSBs. Most differentially expressed ncRNAs were annotated to cell cycle, p53, PI3K-AKT, and WNT signaling pathways. (4) Conclusions: The ceRNA network helps to understand the effects of DNA DSBs activation and remission on the biological function of CCs.

## 1. Introduction

The follicle is a basic component of the mammalian ovary, and its development is coordinated by a variety of factors. Growing follicles develop from primordial follicle, as well as preantral and antral follicle. With the development of the follicle, the oocyte volume gradually increases and, at the same time, accumulates a large amount of genetic material and nutrients. There are multiple layers of cumulus cells (CCs) around it, and the cumulus cell–oocyte complexes (COCs) are close and orderly, which plays an important role in oocyte growth, meiosis, maturation, and ovulation [1,2]. However, most follicles undergo atresia degeneration before ovulation and cannot successfully ovulate in monotocous animals.

COCs are considered functional units of female mammalian germ cells [3]. During follicle development, maintaining genomic integrity is fundamental to ensuring reproductive success. However, long-term disturbances of in vitro and in vivo factors can lead to the accumulation of various lesions, including double-strand breaks (DSBs). DSBs are the most severe form of DNA damage caused by the simultaneous cleavage of double-stranded DNA at opposite or nearby positions by exogenous or endogenous factors [4,5]. γH2AX, formed by phosphorylation at DSB sites, plays an important role in extending the signaling cascade during the DNA damage stress response (DDR), which occurs early after injury and is considered the most effective biomarker for the detection of DSBs [6].

After DSBs occur, if the DDR cannot be properly activated, chromatin remodeling [7], cell cycle arrest [8], apoptosis or other forms of cell death may occur [9]. All of these may cause the quality of oocytes to decline [10], which in turn leads to the failure of oocytes to mature and become fertilized normally, resulting in disease or mutation in the offspring [11,12,13]. Studies have shown that DSBs affect the transition of porcine oocytes from MI to MII, inhibit first polar body excretion and oocyte maturation, and result in MI spindle defects [14,15]. The occurrence of DSBs in bovine oocytes affects their maturation rate [16]. DSBs disrupt bovine COC communication and cause oocytes to exit meiosis [17], and the DNA damage of CCs inhibits oocytes from resuming meiosis and affects blastocyst yield, hatchability, and embryo quality, while bovine embryonic developmental delay is positively correlated with the degree of DNA damage [18,19,20]. CCs control DSB-induced DDR in oocytes [21]. DSBs and premature ovarian failure are associated with the BRCA-AT pathway and the PTEN/PI3K/Akt pathways, whereas the CHK1/CHK2 to p53 and p63 pathways can eliminate oocyte DSBs [22,23,24]. However, research on the reproductive system of female animals mostly focuses on the regulation of coding genes, and research on the regulation of coding genes at the noncoding RNA (ncRNA) level, especially the interaction between ncRNAs, is scarce.

Only approximately 2% of genes in the genome encode proteins, and most of the others encode transcripts that ncRNAs, including long noncoding RNAs (lncRNAs), circular RNAs (circRNAs), and small RNAs (sRNAs) [25,26]. As research has increased, it has been found that ncRNAs are involved in many biological functions and biological processes, including development, proliferation, apoptosis, survival, differentiation, and carcinogenesis [27,28,29,30]. Previous studies have shown that microRNAs (miRNAs) have important roles in bovine oocytes and early embryos. miRNAs also affect blastocyst activation and implantation in rats. The overexpression of miR-155 and miR-224 in mice inhibited CC expansion and oocyte maturation [31,32,33,34]. The knockdown of lncRNA–ROSE results in abnormal oocyte cytokinesis and impaired preimplantation embryo development in mice, and the inhibition of circRNA ARMC4 affects oocyte meiosis and early embryonic development in pigs [35,36]. It is known that the binding of miRNAs to mRNA leads to gene silencing, and competitive endogenous RNAs (ceRNAs) can regulate gene expression by affecting miRNAs through competitive binding to miRNA response elements (MREs), indicating that there are important biologically significant RNA–miRNA regulatory pathways [37]. For example, the lncRNA Lnc-RI, which is a ceRNA, affects DSBs by competitively binding with miR-193a-3p and releasing the expression of RAD51, [38]. circRNA DB activates the USP7/Cyclin A2 signaling pathway through miR-34a, which promotes the increase in USP7 expression and the decrease in DNA damage [39], indicating that the ceRNA regulatory network can affect DSBs in cells. Although research in this field is rapidly advancing, little is known about the impact of ceRNA regulatory networks on DSBs or the DDR. In particular, there is a lack of a systematic analysis of the relationships between DSBs/DDR and ncRNAs in bovine CCs.

Bleomycin (BLM), the fermentation product of Streptomyces verticillus, is a commonly used inducer of cellular DSBs and drug for the treatment of cancer [40]. BLM also affects the cell cycle, proliferation, and apoptosis [41,42]. In this study, bovine CCs were first treated with 200 μM BLM [43] to construct a DSB model. Subsequently, the effects of DSBs on the cell cycle, cell viability, and apoptosis of CCs were detected, and the differential expression profiles of lncRNAs, circRNAs, miRNAs, and mRNAs during the DNA-DSB process of bovine CCs were determined by high-throughput sequencing technology. Finally, we predicted the lncRNA/circRNA–miRNA–RNA regulatory network based on the ceRNA mechanism. The results of this study are helpful for elucidating the mechanism underlying the interaction between ncRNAs and mRNAs in bovine CCs during DSBs/the DDR and provide a meaningful new direction for preventing and improving the effect of DSB-induced oocyte quality reduction on fertility in animals.

## 2. Materials and Methods

### 2.1. CCs Collection, Culture and BLM Treatment

CCs were harvested as we described before [44]. In brief, the ovaries were obtained from healthy cows of Changchun Haoyue Halal Meat Products Co., Ltd., in Jilin Province and transferred to the laboratory within 1 h. Follicular fluid with COC in the follicles 3–8 mm in diameter was obtained with a 10 mL syringe. After washing three times with HEPES (Gibco, Paisley, Scotland, United Kingdom), COCs with more than five tightly wrapped CC layers were collected under a microscope. Subsequently, the CCs were separated from the oocyte with 0.1% hyaluronidase. Then, CCs were harvested and centrifuged for 5 min, washed twice with phosphate-buffered saline (PBS), cultured in DMEM/F12 (Gibco, Grand Island, NY, USA) supplemented with 1% penicillin, streptomycin (HyClone, Logan, UT, USA), and 10% fetal bovine serum (Biological Industries, Kibbutz Beit Haemek, Israel), and placed in an incubator at 38.5 °C under 5% CO_2_. For BLM (Zeocin, Thermo Fisher, Carlsbad, CA, USA, diluted with culture media) treatments, 5 × 10^5^ cells/well CCs were seeded into 6-well cell culture plates and cultured for 6 h. Then, solutions of 0 μM (Negative Control(NC)group) and 200 μM BLM (BLM group) were applied to the cells for 3 h.

### 2.2. γH2AX Detection

After BLM treatments, CCs were dissociated with 500 μL of trypsin for 3 min. After 500× *g* centrifugation and washing three times with PBS, the CCs were fixed in 100 μL of 4% paraformaldehyde solution for 10 min. Then, 900 μL of prechilled methanol was added, after which the mixture was gently homogenized and then incubated on ice for 30 min. Next, the cells were centrifuged at 500× *g* for 5 min, transferred to 1 mL of incubation solution (500 g bovine serum albumin in 100 mL PBS) and mixed well after 500× *g* centrifugation. CCs were treated with 100 μL of incubation solution-diluted (1:50) Alexa Fluor 488-labeled γH2AX antibody (Cell Signaling Technology, Danvers, MA, USA) for 1 h at room temperature. Subsequently, the cells were washed three times with incubation solution and then centrifuged at 500× *g* for 5 min. Finally, the cells were resuspended in 200 μL of PBS and analyzed using flow cytometry (Beckman Coulter, Brea, CA, USA).

### 2.3. Cell Cycle Analysis

The cell cycle was determined in strict accordance with the instructions of the cell cycle kit (Beyotime, Shanghai, China). CCs (5 × 10^5^ cells/well) were seeded in 6-well plates. After the cells adhered, they were treated with 0 μM BLM-NC or 200 μM BLM for 3 h, trypsinized, and collected into 1.5 mL centrifuge tubes. Cells were resuspended in 100 μL of PBS after centrifugation at 500× *g* for 5 min. The cell cycle distribution was detected using a flow cytometer (Beckman Coulter, Brea, CA, USA), and the data were processed using MODFIT software (Verity Software House, Topsham, ME, USA).

### 2.4. Cell Proliferation Assays

The proliferation of bovine CCs was measured using a CCK-8 kit (Dojindo, Kyushu, Japan). In brief, each well of a 96-well plate contained 1 × 10^4^ CCs in 100 μL of media and was cultured for 12 h at 38.5 °C and 5% CO_2_. BLM (0 μM) was added to the NC group, or BLM (200 μM) was added to the BLM group. The normal media was replaced immediately after 3 h of treatment. After 0, 24, 48, and 72 h of incubation, 10 μL of CCK-8 solution was added to each well. The plates were incubated in a 38.5 °C incubator for 3 h, and the absorbance at 450 nm was measured using a microplate reader (BioTek Instruments, Winooski, VT, USA).

### 2.5. Apoptosis Analysis

The method of detecting apoptosis was performed strictly in accordance with the apoptosis assay kit instructions (BD, CA, USA). Bovine CCs (5 × 10^5^ cells/well) were seeded into 6-well plates. Cells were treated with BLM (0 or 200 μM) for 3 h. Then, 100 μL of PBS was added to each centrifuge tube, followed by the addition of 5 μL of FITC solution and 5 μL of propidium iodide (20 μg/mL). The cells were subsequently incubated for 15 min at room temperature in the dark. The samples were measured within 2 h using a flow cytometer (Beckman Coulter, Brea, CA, USA). Apoptotic cells and dead cells were distinguished by staining with propidium iodide and FITC.

### 2.6. RNA Extraction

After treatment, total RNA was extracted using TriPure (Roche, Basel, Switzerland), according to the manufacturer’s recommended protocol. All the RNA samples were checked using a NanoDrop ND-2000 Spectrophotometer (Thermo, Waltham, MA, USA), a Qubit 2.0 instrument (Thermo, Waltham, MA, USA), and an Agilent 2100 device (Agilent, Karlsruhe, Baden-Württemberg, Germany) and via electrophoresis and were then stored at −80 °C for further experiments.

### 2.7. LncRNA, circRNA, and mRNA Library Construction

An Epicenter Ribo-ZeroTM kit (Epicenter, Madison, WI, USA) was used to remove rRNA from the sample, and then fragmentation buffer was added to randomly interrupt rRNA-depleted RNA. First-strand cDNA was synthesized with six-base random primers, and then buffer, dATP, dUTP, dCTP, dGTP, RNase H, and DNA polymerase I were added to synthesize doubled-stranded cDNA. This cDNA was purified using AMPure XP magnetic beads, the purified double-stranded cDNA was end repaired, A was added, and the sequencing adapter was ligated. Then, AMPure XP magnetic beads were used for fragment size selection. Finally, the U chain was degraded, and the cDNA library was obtained via PCR enrichment. LncRNA and circRNA libraries were sequenced by the Illumina HiSeq platform (Biomarker Technologies, Beijing, China).

### 2.8. miRNA Library Construction

Six microliters of RNase-free ddH2O solution, including 1.5 μg of RNA, was used to construct a library by using a Small RNA Sample Pre Kit (Thermo, Waltham, MA, USA). T4 RNA Ligase 1 and T4 RNA Ligase 2 (truncated) were attached to the 3′ and 5′ ends of the small RNA due to the phosphate group at the 5′ end and the hydroxyl group at the 3′ end, respectively. cDNA, PCR amplification, and gel separation techniques were used to screen the target fragments, and the fragments obtained by gel extraction were considered small RNA libraries.

After the library was constructed, a Qubit 2.0 instrument (Thermo, Waltham, MA, USA) was used to test the library concentration, the library concentration was diluted to 1 ng/μL, and an Agilent 2100 Bioanalyzer (Agilent, German) was used to detect the insert size. qPCR was used to accurately quantify the effective concentration of the library to ensure the quality of the library. The miRNA library was sequenced using HiSeq 2500 high-throughput sequencing (Biomarker Technologies, Beijing, China).

### 2.9. Bioinformatic Analysis of LncRNA

Transcriptomes were assembled using StringTie based on data mapped to the reference genome. Assembled transcripts were annotated using the gffcompare program (The Center for Computational Biology, USA). The CPC/CNCI/CPA/Pfam programs were combined to predict nonprotein-coding RNAs from unknown transcripts [45,46,47,48]. Transcripts that were more than 200 nt in length and with more than two exons were selected as lncRNA candidates, and CPC/CNCI/CPAT/Pfam with the ability to distinguish protein-coding genes from noncoding genes was used to further screen different types of lncRNAs, including lincRNAs, intronic lncRNAs, antisense lncRNAs, and sense lncRNAs.

### 2.10. Bioinformatic Analysis of circRNAs

CIRI and find_circ software [49] were used to predict circRNAs. First, CIRI software was used to predict circRNA sequences. Then, the find_circ software first took 20 bp from both ends of the reads on the genome alignment as anchor points and then used the anchor points as independent reads on the genome to align and identify the only matching site. If the alignment positions of the two anchors are reversed in the linear direction, the reads of the anchors are extended until the joint position of the circular RNA is found. A GT/AG splicing signal is considered to indicate circRNA.

### 2.11. Bioinformatic Analysis of miRNA

We obtained clean data from the raw data by removing reads containing adapters, reads containing poly-N sequences, and reads of low quality, trimmed and cleaned the data by removing sequences smaller than 18 nt or larger than 30 nt, and calculated the Q20 value, Q30 value, GC content, and sequence repeat levels. By the use of Bowtie tools [50], the clean reads were compared with the content in the Silva database, GtRNAdb database, Rfam database, and Repbase database to filter ribosomal RNA (rRNA), transfer RNA (tRNA), small nuclear RNA (snRNA), small nucleolar RNA (snoRNA), and other ncRNAs and repeats. Next, the remaining reads were used to detect known miRNAs, and newly discovered miRNAs were obtained by comparison with miRBase. Novel miRNA secondary structures were predicted using MTide tools [51].

### 2.12. Expression Analysis

The DESeq R package (1.10.1) [52] was used to analyze the expression of RNAs in each sample, and a fold change (fold change) greater than or equal to 1.5 and an FDR less than 0.05 were used as the criteria for screening differentially expressed RNAs. A fold change represents the ratio of expression levels between the two groups of samples.

### 2.13. Functional Enrichment Analysis

We used DAVID software to analyze the Gene Ontology (GO) of differentially expressed RNAs [53,54]. The Kyoto encyclopedia of genes and genomes (KEGG) was used to analyze differentially expressed RNAs with the KOBAS software [55]. GO and KEGG pathways identified as having a *p* value < 0.05 were considered significantly enriched.

### 2.14. RT–qPCR

The differential expression levels of ncRNAs and mRNAs in the NC group or BLM group were measured by RT–qPCR. To determine the resistance of ncRNAs to RNase R digestion, total RNA was treated with RNase R (Epicenter, WI, USA) prior to cDNA synthesis. Fluorescence quantitative PCR was performed using SYBR Green (Tiangen, Beijing, China) according to the manufacturer’s protocol. Each 20 μL of the RT–qPCR mixture consisted of 8 μL of deionized water, 10 μL of SuperReal PreMix Plus (Tiangen, Beijing, China), 1 μL of cDNA, and 0.5 μL each of forward and reverse primers (10 mM). The RT–qPCR conditions included denaturation at 95 °C for 180 s and 40 cycles of 95 °C for 15 s, 60 °C for 20 s, and 72 °C for 15 s. Gene expression was quantified using a Mastercycler ep realplex (Eppendorf, Hamburg, Germany) and the 2^-ΔΔct^ method with GAPDH or miRNA-U6 as the standard. The sequences of the primers used are shown in Appendix A.

### 2.15. CeRNA Regulatory Network Analysis

The miRanda, RNAhybrid, and TargetScan databases were used to predict the miRNAs targeting mRNAs, lncRNAs, and circRNAs. The input files were miRNA, mRNA, lncRNA, and circRNA base sequence files that we obtained from the RNA-seq. Then, the miRNA, mRNA, lncRNA, and circRNA with different expressions were selected to construct the ceRNA network. Overlapping the same miRNAs to construct a lncRNA/circRNA–miRNA–mRNA regulatory network and the interaction between the networks were analyzed using Cytoscape software [56]. The sequences of the ncRNAs used are shown in Appendix A.

### 2.16. Statistical Analysis

The experimental results are expressed as the means ± SDs of three independent experiments. The data were analyzed using the ANOVA program of SPSS 23.0 (IBM, Armonk, NY, USA), and *p* < 0.05 was considered significant.

## 3. Results

### 3.1. BLM Treatment Induced DSBs, Disrupted the Cell Cycle, and Inhibited Cell Proliferation in Bovine CCs

As shown in Figure 1a, the γH2AX-positive rate of CCs in the BLM group was higher than that in the NC group (*p* < 0.01). The results (Figure 1b) showed that the relative cell cycle level in the G1 phase of the BLM group was lower than that of the NC group (*p* < 0.01), while that of the S phase was higher than that of the NC group (*p* < 0.01). In the G2 phase, the relative cell cycle levels of the two groups were not different (*p* > 0.05). The cell cycle of bovine CCs treated with BLM was altered, the retention time of the G1 phase in the BLM group was significantly lower than that of the control group, and the retention time of the S phase was significantly higher than that of the control group.

CCK8-related data (Figure 1c) showed that the cell viability increased in both the NC and the BLM groups, but the cell viability in the BLM group was lower than that in the NC group at 3 + 0 h, 3 + 24 h, 3 + 48 h, and 3 + 72 h (*p* < 0.01).

In addition, the relative apoptosis of CCs induced by BLM was higher than that of the NC group (Figure 1d), but the difference was not significant. This indicated that most CCs treated with BLM did not begin the apoptosis process.

### 3.2. Overview of RNA Sequencing

We constructed two cDNA libraries, each with three replicates, to identify the mRNA, lncRNA, and circRNA characteristics and abundances of CCs in the NC group and the BLM group. We obtained a total of 124.06 Gb of clean data for mRNA, lncRNA, and circRNA analysis, and the clean data per sample reached 17.75 Gb. After removing junction and low-quality reads, 143,882,865 and 132,777,966 clean reads were obtained from the NC and BLM groups, respectively. The average GC content in the NC group and BLM group was 56.58% and 51.74%, respectively. With reference to the bovine genome sequence (NCBI: txid9913), we obtained 95.54% and 83.79% of the mapped reads of the clean data from the NC group and the BLM group, respectively. The sequencing depth of the NC group was 7.4345 ± 0.3171, and that of the BLM group was 8.3795 ± 0.5218. Appendix A shows the distribution of mapped reads covering depth on the reference genome. The Q30 value of the NC group was 94.39% ± 0.1955%, and that of the BLM group was 94.52% ± 0.1169%.

We also obtained a clean data of 147.98 M for miRNA analysis. We also performed some processing of the clean read, such as removing low-quality sequences, reads with unknown base content greater than or equal to 10%, reads without 3′ linker sequences, 3′ linker sequences, and sequences that were less than 18 or more than 30 nucleotides in length; we ultimately obtained 25,735,570 and 17,420,398 clean reads from the NC and BLM groups, respectively. We referenced the bovine genome sequence (NCBI: txid9913) and obtained 70.91% and 69.20% mapped reads from the NC and BLM groups, respectively. The distribution of the mapped reads coverage depth across the reference genome is shown in Appendix A. The Q30 values of the NC group and BLM group were 98.91% ± 0.0698%, and 98.88 ± 0.0458, respectively.

### 3.3. Differentially Expressed mRNA Identification

After the bovine genome reference sequences were compared, 1750 (3275 in total) functional mRNAs were annotated. The length of mRNA was mainly concentrated at 400 bp or more than 3000 bp (Figure 2a). From the mRNA expression profile data, 848 mRNAs were differentially expressed (fold change ≥ 1.5, *p* < 0.05), of which 396 mRNAs were upregulated and 452 were downregulated and showed hierarchical clustering (Appendix A). The mRNAs listed in Appendix A have been shown to be associated with DSBs. Next, three upregulated and three downregulated mRNAs were randomly selected to verify the sequencing data by RT–qPCR. The results showed that they have the same expression pattern as that in the sequencing data (Figure 2b).

Subsequently, we performed GO term and KEGG pathway analyses on all differentially expressed mRNAs. The results showed that GO terms (Appendix A) mainly focused on cellular processes, biological regulation, metabolic processes, and responses to stimuli, which were associated with DSBs and the DDR. The KEGG pathway (Figure 2c) analysis revealed pathways related to the cell cycle, p53 signaling, and PI3K-Akt signaling. This indicated that BLM-induced DSBs in CCs would cause the differential expression of mRNA and lead to changes in the biological state of cells.

### 3.4. Differentially Expressed lncRNA Identification

The analysis of CNCI, CPC, Pfam, and CPAT data revealed 6436 lncRNAs: 3388 lincRNAs (52.6%), 886 antisense lncRNAs (13.8%), 1720 intron lncRNAs (26.7%), and 442 lncRNAs (6.9%) (Figure 3a,b). The predominant length of differentially expressed lncRNAs was between 200 and 800 bp (Figure 3c). Furthermore, the expression levels of protein-coding genes were slightly higher than those of lncRNA transcripts (Figure 3d).

Among the 75 differentially expressed lncRNAs, 38 were upregulated, while 37 were downregulated (Appendix A). Three upregulated and three downregulated lncRNAs were randomly selected for RT–qPCR, and their expression patterns (Figure 4a) were consistent with the sequencing data.

Next, the identified GO terms and KEGG pathways were used to explore the potential functions of lncRNAs by searching 100 kb protein-coding gene sequences upstream and downstream of all identified lncRNAs to predict potential cis-regulatory targets of lncRNAs. A GO enrichment analysis showed that these genes were mainly related to GO terms involving cellular processes, cellular components, and binding (Appendix A). As shown in Figure 4b, the pathways significantly enriched by KEGG mainly included the Hippo signaling pathway, the Wnt signaling pathway, and pathways related to the adherens junction and tight junction.

### 3.5. Differentially Expressed circRNA Identification

A total of 594 circRNAs, including exons, intergenic regions, and introns, which were mainly 400–800 bp in length, were identified (Figure 5a). Among them, the number of intronic circRNAs was the largest, followed by intergenic regions, and exons were the least abundant (Figure 5a). Thirty-two were upregulated, while 34 were downregulated (Appendix A) (fold change ≥ 1.5, *p* < 0.05). Three upregulated and three downregulated circRNAs were randomly selected for RT–qPCR (Figure 5b), and the results were consistent with the HiSeq data.

The GO analysis revealed that many host genes were closely related to biological processes, cellular components, and molecular functions (Appendix A), including cellular processes, cells, cellular parts, binding, and catalytic activities. Different gene products coordinate with each other to perform biological functions, and the pathway annotation of circRNA host genes can better help to understand the functions of these genes. KEGG analysis showed that 15 pathways, including those related to NF-κB, mRNA surveillance, and lysine degradation, were associated with potential CC functions (Figure 5c).

### 3.6. Differentially Expressed miRNA Identification

In this study, 1836 miRNAs, of which 467 were known and 1369 were newly predicted, were identified. Most miRNAs were mainly between 20 and 24 bp in length (Figure 6a). The results showed (Appendix A) 40 upregulated miRNAs and 31 downregulated miRNAs (fold change ≥ 1.5, *p* < 0.05). Among the 15 known differentially expressed miRNAs, three are involved in DSBs, including miR-486, miR-451, and miR-145 [57,58,59] (Appendix A). Then, six randomly selected miRNAs were verified by RT–qPCR (Figure 6b), and their levels were consistent with the sequencing data. We analyzed the nucleotide bias at each position of the miRNA (Figure 6c).

Next, we predicted the functions of differentially expressed miRNAs by targeting relationships between miRNAs and mRNAs, as well as performing further GO (Appendix A) and KEGG (Appendix A) analyses. Cellular processes, single biological processes, metabolic regulation, cellular fractions, cells, binding, etc., were the main components identified via the GO analysis. The KEGG functional enrichment pathway analysis mainly revealed pathways involved in endocytosis, lysosome, protein processing, axon guidance, transcriptional dysregulation in cancer, dilated cardiomyopathy, neuroligand–receptor interaction, etc.

### 3.7. Construction of the lncRNA–miRNA–mRNA Regulatory Network

We predicted the ceRNA regulatory network of lncRNA–miRNA–mRNA using Cytoscape software (Figure 7). Figure 7a shows 52 groups of lncRNA–miRNA–mRNA regulatory networks, including lnc-MSTRG.197583.1, which acts as a molecular sponge to adsorb miR-unconservative-AC-00017.1-311957 to promote the expression of mRNAs. lnc-MSTRG.69684.1, 155681.4, and 24637.1 interact with miR-unconservative-AC-000166.1-251601, resulting in abnormal mRNA expression, and lnc-MSTRG.177599.1 affects miR-unconservative-AC-000173.1-360722 and 000162.1-112908 to target the same mRNA. The MiRNA-mediated regulation of NOTCH1 and CEP170B by lnc-MSTRG.197583.1, 69684.1, 155681.4, and 24637.1, lnc-MSTRG.177599.1, 69684.1, 155681.4, and 24637.1 regulate CEP170B and SPTBN5.

In addition, five repressed lncRNAs promoted the expression of five miRNAs and decreased the expression of 24 mRNAs consisting of 26 ceRNAs (Figure 7b), such as lnc-MSTRG.146080.1, 49414.1, 112314.1, 7914.1, and 16, which promote miR-unconservative-AC-000178.1-450303, 000176.1-413881, 000176.1-409242, 000169.1-288145, and 000163.1, respectively. Lnc-MSTRG.146080.1 and 49414.1 allowed the expression of targeted miRNAs and corepressed MDM1 and PAXIP1.

### 3.8. Construction of the circRNA–miRNA–mRNA Regulatory Network

We predicted a ceRNA regulatory network involving 276 circRNA–miRNA–mRNA pairs using Cytoscape software, of which 124 pairs promoted the upregulation of mRNA expression (Figure 8a) and 151 pairs inhibited mRNA expression (Figure 8b). When circRNA–AC_000176.1: 39742140|39768649, AC_000175.1: 46155528|46202870, and AC_000158.1: 107948998|107975652 jointly suppressed miR-unconservative_AC_000159.1_32276, unconservative_AC_000162.1_112908, AC_000171.1_312038, and AC_000173.1_360722 released 18 mRNAs. CircRNA-AC_000159.1:20992753|21047706 acted as a molecular sponge to adsorb miR-unconservative_AC_000162.1_112908 and AC_000173.1_360722 to increase mRNA expression.

### 3.9. Construction of the LncRNA/CircRNA–miRNA–mRNA Regulatory Network

We predicted five groups of lncRNA/circRNA–miRNA–mRNA regulatory networks based on the same miRNA-binding sites (Appendix A). For example, the joint regulation of mRNA by lncRNA MSTRG.197583.1 and circRNA AC_000176.1:39742140|39768649 and AC_000182.1:38287754|38329596 via miR-unconservative_AC_000171.1_311957, miR-unconservative_AC_000162.1_112908, and AC_000173.1_360722 mediated lncRNA MSTRG.177599.1 and circRNA AC_000158.1: 107948998|107975652, AC_000159.1: 20992753|21047706, AC_000171.1: 63954121|64038908, and AC_000175.1: 46155528|46202870 promoted mRNA expression; miR-unconservative_AC_000168.1_251602 was inhibited by lnc-MSTRG.155681.4, MSTRG.24637.1, MSTRG.69684.1, and circ-AC_000171.1 Furthermore, the inhibition of lnc-MSTRG.112314.1 and circ-AC_000181.1:934078|947669 released the expression of miR-unconservative_AC_000176.1_413881 to silence the mRNA (Appendix A). These results showed that the treatment of CCs with BLM resulted in changes in the expression levels of various components, and a series of multiple regulatory events took place along with them.

## 4. Discussion

BLM has a dose/time-dependent effect on the induction of γH2AX, which is formed by DSBs through the generation of free radicals [15,60]. The formation of γH2AX functions not only in DSBs, but also as a recruitment factor for the DDR [61]. CCs regulate oocyte development through gap links, and DSBs cause bovine COC communication disorder, ultimately causing oocytes to exit meiosis [17] and control the DSB-induced oocyte DDR process [21], this leads to DNA injury and inhibits the oocyte resumption of meiosis, which generates DNA damage that inhibits the recovery of oocyte meiosis [18]. DNA damage to cells leads to cell cycle arrest [62], inhibiting cell viability [63], but cells complete DNA replication and histone synthesis in S phase to lay the foundation for cell division [64]. These results indicated that after BLM induced DSBs in CCs, a large amount of γH2AX was formed, which altered the cell cycle and cell viability and then affected the development of oocytes. Our study showed (Figure 1) that after bovine CCs were treated with 200 μM BLM for 3 h, the level of γH2AX in CCs increased, which promoted the rapid passage of cells through the G1 phase, but a large number of cells were arrested in the S phase, which in turn affected cell proliferation but did not significantly affect apoptosis. These results suggest that DSBs participate in oocyte development by regulating the biological functions of CCs. However, the molecular mechanism is still unclear, especially the relationship between ncRNA and DSB. Therefore, we performed high-throughput sequencing of the DSB model of CCs to investigate the potential functions of mRNA, lncRNA, circRNA, and miRNA during DSB/DDR processes in CCs.

During double strand breaking, cells are bound to detect damage and temporarily block cell cycle progression to allow time to repair or exit the cell cycle [65]. The rest of the cell cycle is required to engage the DDR through the cell cycle checkpoint pathway in response to DSBs [66]. Our KEGG pathway analysis (Figure 2c and Figure 4b) showed that the most enriched cellular process was the cell cycle. This suggests that cell cycle arrest caused by DSBs in CCs may promote DDR to maintain the biological function of CCs and ensure the connection between CCs and oocytes. Our analysis also showed that KEGG also enriched the p53 and PI3K-AKT signaling pathways (Figure 2c and Figure 4b), which play very important roles in DSB and DDR processes. On the one hand, DNA damage induces ATM lysine acetylation to activate ATM. The kinase activity, in turn, activates the ATM-chk2-p53 pathway and the PI3K-AKT pathway [67,68]. The p53 pathway can eliminate DSBs in oocytes [24], and the PI3K-AKT pathway promotes DSB repair [69]. Therefore, our experiments show that after the occurrence of DSBs in bovine CCs, the main purpose of the cell cycle, p53 signaling pathway, and PI3K-AKT pathway is to activate the DDR and eliminate DSBs in cells to stabilize the genome and maintain the normal function of CCs and oocytes. However, there are still many pathways related to CCs or oocytes, DSBs/the DDR that have not been investigated. We will investigate this in future research. In addition, we need to explore whether DSBs/the DDR is caused by pathway alteration or whether DSBs activate the pathway.

Currently, the analysis of DSBs is mainly related to genes encoding proteins, while little is known about ncRNAs. Recent reports have demonstrated that ncRNAs are involved in the regulation of multiple biological functions and pathological processes by regulating gene expression at the transcriptional and posttranscriptional levels [70], such as aging [71]. In addition, the competitive binding of lncRNA RI to miR-193a-3p affects homologous recombination(HR) repair of DSBs [38]. CircRNA circ-DB affects DNA damage through the regulation of USP7 by miR-34a [39]. LncRNAs regulate the cell cycle and proliferation through p53 [72,73]. miRNAs affect the cell cycle or apoptosis through p53 [74,75,76,77,78], and the p53 pathway is closely related to oocyte DSBs [24]. This finding indicates that ncRNA can affect DSBs of oocytes through the p53 signaling pathway. We will investigate this in future research. In this study, we discovered 6436 lncRNAs, 594 circRNAs, and 1836 miRNAs from the DSB model of bovine CCs using the Illumina HiSeq Xten platform. It is well known that CCs play an important role in the maturation of oocytes. To our knowledge, this study is the first to determine the expression pattern of ncRNAs following the induction of DSBs in bovine CCs.

Although less conservative and expressed at lower levels than protein-coding genes, ncRNAs are often regulated by transcription factors and expressed specifically in cells [29]. It has been reported that lncRNAs can act as miRNA sponges to regulate target genes [38]. FBXW7 can be rapidly recruited to DNA damage sites, and its phosphorylation mediated by ATM promotes its retention at DNA damage sites for subsequent NHEJ repair [79]. lncRNA–MIF inhibits tumorigenesis through miR-586 and attenuates the inhibitory effect of miR-586 on FBXW7 [80]. The lncRNA TINCR, as a ceRNA, cleaves miR-544a from its target gene FBXW7 to regulate the proliferation and invasion of lung cancer cells [81]. DNA damage activates PIDD1 and NF-κB to promote cell survival [82], and the loss of PAXIP1 promotes DDR [83]. RFWD3-mediated ubiquitination promotes the HR [84]. In this study, 38 lncRNAs were upregulated and 37 lncRNAs were downregulated, and some lncRNAs were predicted to be involved in the ceRNA regulatory network, suggesting that lncRNAs may be involved in the DSB/DDR regulatory process. Based on previous studies, we speculate that lnc-MSTRG.197583.1 may act as a sponge to adsorb miRNAs to release FBXW7 to promote the DDR. After DSBs occur, miR-unconservative-AC-000166.1-251601 may be inhibited by lnc-MSTRG.69684.1, 155681.4, and 24637.1 to promote PIDD1 mRNA expression, thereby maintaining cell stability. The deletion of lnc-MSTRG.146080.1 releases miRNA, which inhibits the expression of PAXIP1 indirectly, possibly promoting the DDR. lnc-MST·RG.197668.1 may regulate the efficiency of RFWD3 HR mediated by miRNA. These findings indicate that BLM induces DSBs in CCs, resulting in changes in the expression of a large number of ncRNAs. They affect the γH2AX, cell cycle, and cell viability of CCs through regulatory networks, thereby affecting the development and biological functions of oocytes, and even follicles. In bovine CCs, the relationship between most ceRNA regulatory networks and DSBs/the DDR is unclear, warranting experimental validation based on the predicted results.

DDR is an important defense mechanism against genomic instability [85]. The presence of DSBs in cells activates the DDR, which is an extensive signaling network that includes DNA repair, cell cycle checkpoint activation, cellular senescence, and apoptosis [86]. DDR involves several pathways including HR and NHEJ to repair DSBs [87]. Loss of function and mutation of key DDR genes lead to premature ovarian failure [88]. Non-coding RNA (ncRNA) has recently emerged as a vital component of the DNA damage response (DDR), which was previously believed to be solely regulated by proteins. Many species of ncRNA can directly or indirectly influence DDR and enhance DNA repair, particularly in response to double-strand DNA breaks. The abnormal expression of miRNA can affect the DDR [89]. miR-34a regulates DDR through FOXP1 [90]. lncRNAs control the DNA damage response through interaction with DDRNAs at DSBs [91]. LncRNA DDSR1 has both an early role by modulating repair pathway choices, and a later function when it regulates gene expression [92]. Circ_0057504 promotes DDR via the NONO-SFPQ complex [93]. The LncRNA OTUD6B-AS1/miR-26a-5p/MTDH pathway affects the stability of DDR genes [94]. By overlapping the same miRNAs, we predicted the lncRNA/circRNA–miRNA–mRNA regulatory network (Appendix A) and found a total of five groups. In these regulatory networks, the differential expression of lncRNA/circRNA and mRNA was negatively correlated with miRNAs, which may be because the abnormal expression of lncRNA/circRNA caused by DSBs affects the role of miRNA-induced mRNAs in cell survival or DDRs, or the abnormal expression of these RNAs results in DSBs/the DDR. These changes will have effects on CCs, which in turn affect oocyte development. However, these are only guesses based on previous research reports combined with our findings, and further experimental verification is needed.

## 5. Conclusions

In conclusion, we systematically investigated the potential interactions between ncRNAs and mRNAs in the DSB/DDR processes of bovine CCs. For example, lnc-MSTRG.197583.1 may act as a sponge to adsorb miRNAs to release FBXW7, miR-unconservative-AC-000166.1-251601 may be inhibited by lnc-MSTRG.69684.1, 155681.4, and 24637.1 to promote PIDD1 mRNA expression, the deletion of lnc-MSTRG.146080.1 releases miRNA, which inhibits the expression of PAXIP1 indirectly, and lnc-MST·RG.197668.1 may regulate RFWD3 mediated by miRNAs. These findings will provide insight into the DNA-DSB process and its ability to mediate follicular development and atresia.

## Figures and Tables

**Figure 1 genes-14-00290-f001:**
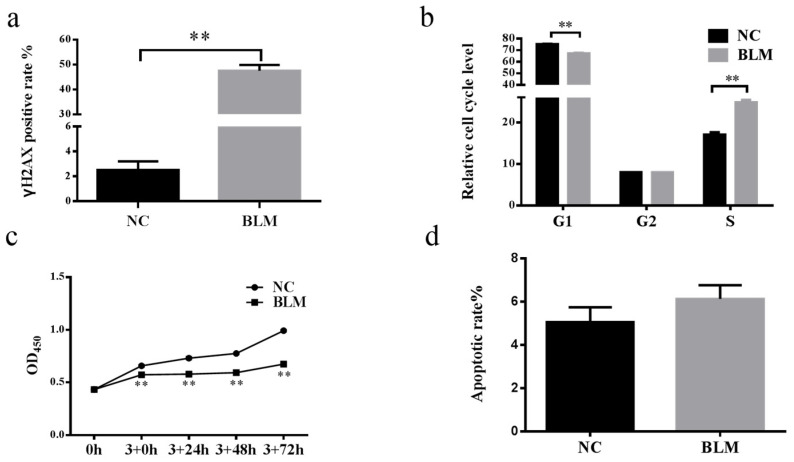
Effect of BLM on bovine CCs. Bovine CCs were treated with BLM (200 μM) for 3 h. (**a**) BLM treatment enhanced the γH2AX-positive rate. (**b**) The percentage of S phase was increased, and G1 phase was decreased significantly. (**c**) The value of OD450 was undermined after treatment with BLM for 3 h and measured for 72 h at intervals of 24 h. (**d**) BLM did not affect the cell apoptosis rate. Significant differences are represented with ** (*p* < 0.01).

**Figure 2 genes-14-00290-f002:**
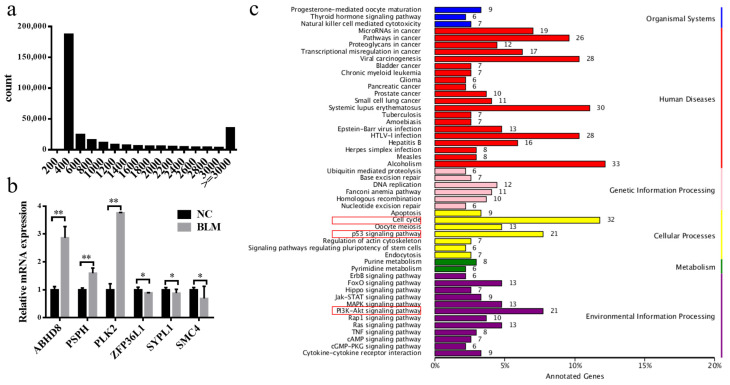
Identification of differentially expressed mRNAs. (**a**) MRNA length analysis. (**b**) Expression of up- and downregulated mRNAs in the NC or BLM group, as determined by RT–PCR. (**c**) KEGG pathway analysis of differentially expressed mRNAs. Red box is a relevant path for highlight DSBs or DDR. Significant differences are represented with * (*p* < 0.05) and ** (*p* < 0.01).

**Figure 3 genes-14-00290-f003:**
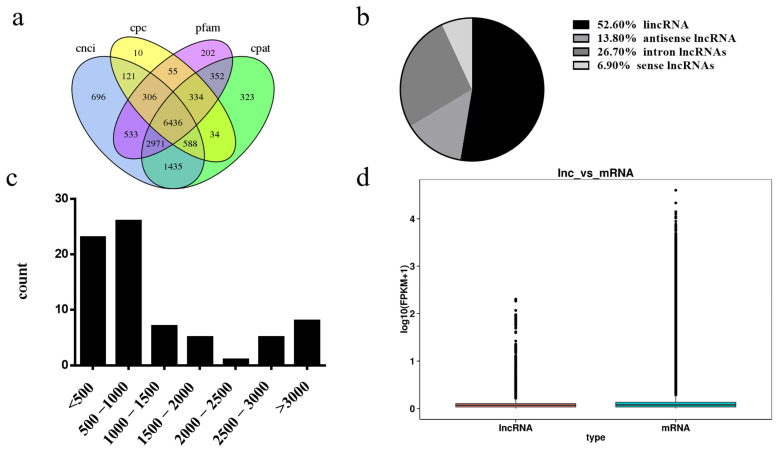
Analysis of lncRNAs. (**a**) Cross-analysis of CNC, CPC, Pfam, and CPAT data revealing lncRNAs. (**b**) Classification of 6436 lncRNAs, including lincRNAs, antisense lncRNAs, intronic lncRNAs, and sense lncRNAs. (**c**) Distributions of the sequence length of lncRNAs. (**d**) Boxplots of the expression levels (log10(FPKM)) of lncRNAs and protein-coding genes.

**Figure 4 genes-14-00290-f004:**
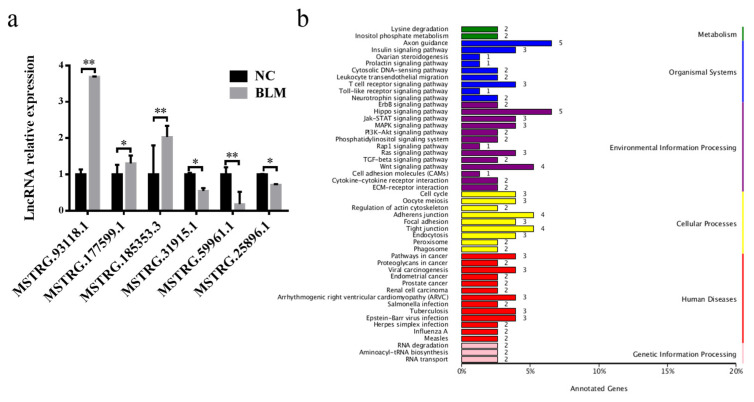
Identification of differentially expressed lncRNAs. (**a**) Expression of up- and downregulated lncRNAs in the NC and BLM groups, as determined by RT–PCR. (**b**) KEGG pathway analysis of differentially expressed lncRNAs. Significant differences are represented with * (*p* < 0.05) and ** (*p* < 0.01).

**Figure 5 genes-14-00290-f005:**
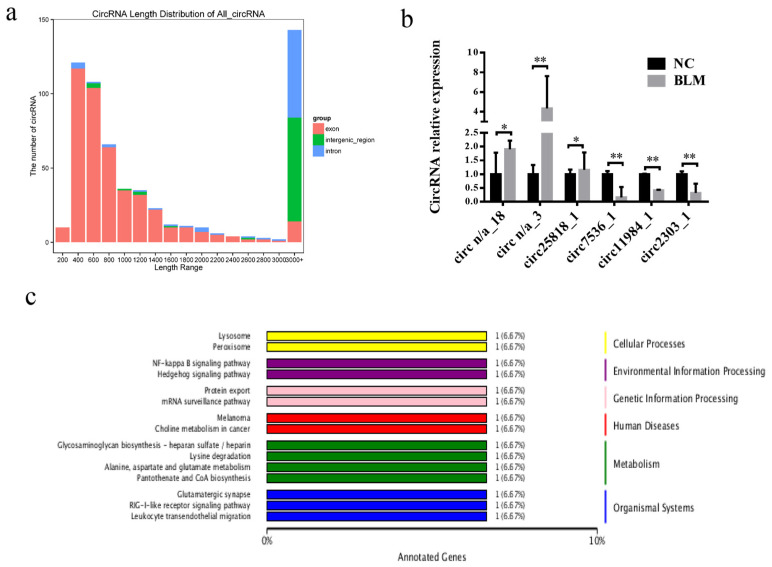
Identification of differentially expressed circRNAs. (**a**) CircRNA length and derivation analysis. (**b**) Expression of up- and downregulated circRNAs in the NC and BLM groups, as determined by RT–PCR. (**c**) KEGG pathway of differentially expressed circRNAs. Significant differences are represented with * (*p* < 0.05) and ** (*p* < 0.01).

**Figure 6 genes-14-00290-f006:**
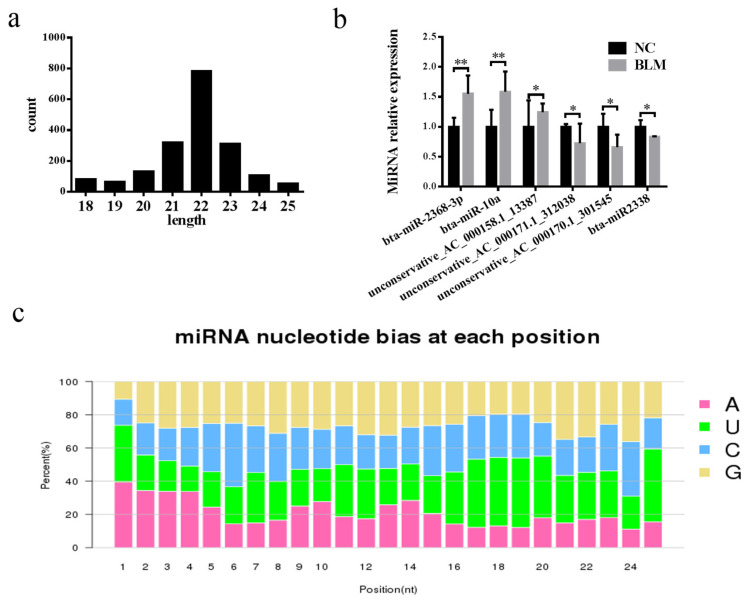
Identification of differentially expressed miRNAs. (**a**) miRNA length analysis. (**b**) Expression of up- and downregulated miRNAs in the NC and BLM groups, as determined by RT–PCR. (**c**) miRNA nucleotide bias at each position. Significant differences are represented with * (*p* < 0.05) and ** (*p* < 0.01).

**Figure 7 genes-14-00290-f007:**
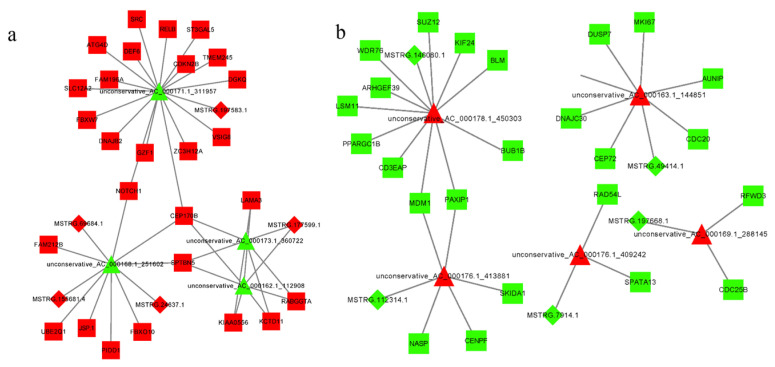
Construction of the lncRNA–miRNA–mRNA ceRNA regulatory network. (**a**,**b**) lncRNA–miRNA–mRNA ceRNA regulatory network. The diamonds represent lncRNAs, the triangles represent miRNAs, the squares represent mRNAs, red indicates upregulation, and green indicates downregulation.

**Figure 8 genes-14-00290-f008:**
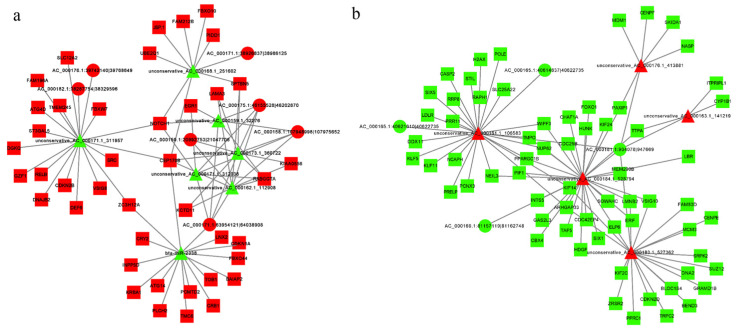
Construction of the circRNA–miRNA–mRNA ceRNA regulatory network. (**a**,**b**) circles represent circRNAs, triangles represent miRNAs, squares represent mRNAs, red indicates upregulation, and green indicates downregulation.

## Data Availability

The data used to support the findings of this study are available from the corresponding author upon request.

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
