# Peer review of "DNA Double-Strand Break-Related Competitive Endogenous RNA Network of Noncoding RNA in Bovine Cumulus Cells"

_genes, 2023, doi:10.3390/genes14020290_

Round 1

Reviewer 1 Report

In this study, the authors try to understand the link between Double Strand Breaks (DSBs) and RNA expression profiles in CCCs. The authors have identified a myriad of RNA species that have altered expression levels and have tried to build pathways based on the hits identified. While the results are interesting, the study lacks mechanistic insight on how the changes in the expression levels of the different RNA species impacts DSB repair and genomic integrity.

The main concerns are:

·      The authors do not explain why using this model system is a good idea for the current study. Would these results corroborate in another system?

·      Line 279-280: The authors mention the effect of BLM but they do not corroborate it with previous literature. Dose their effect agree with previous published work?

·      Line 297-298: The authors mention how many GBs of data they have obtained and how much it reduces after “clean-up”. I am not sure that is necessary or relevant as a yardstick of the quality of their data. Rather , it would be useful to know the coverage depth of sequencing and other QC measures.

·      In Fig 2c, they mention DDR pathways were identified. This should be highlighted in the plot.

·      In section 3.9, the authors just list a number of RNA species identification numbers, but the function/relevance of the particular RNA species does not resonate with the reader. It would be useful to highlight key DDR genes that are known to be regulated by the RNA species identified.

·      Since the authors suggest that the altered transcription profiles leads to changes in DDR responses, it would be good to see some experimental validation of some key genes.

·      The authors do not provide any mechanistic insights into how the altered transcription levels affect DDR in the cells. It would be imperative to have a paragraph in the discussion explaining possible mechanisms for this phenomenon.

Minor comments:

·      The abstract needs to be reworked to bring out the significance of the study

·      Font size in all figures to small, labels illegible

·      Line 21: “these physiological processes”: which process-damage or repair?

·      Line 32: They state that ceRNA help understand the effects of DNA DSB activation. How this is done is unclear. Presumably, the crRNA do not help understand the effect but help in DSB repair?

·      In Fig 1b, there are no error bars and statistics on the “G2” bars in the graph

·      In Fig 1d, there are no statistics shown in the graph

Author Response

Reviewer #1

Reviewer #1: In this study, the authors try to understand the link between Double Strand Breaks (DSBs) and RNA expression profiles in CCCs. The authors have identified a myriad of RNA species that have altered expression levels and have tried to build pathways based on the hits identified. While the results are interesting, the study lacks mechanistic insight on how the changes in the expression levels of the different RNA species impacts DSB repair and genomic integrity.

Response:Thanks for your comments and suggestions. The purpose of this study was to predict the ceRNA regulatory network caused by DSB in bovine cumulus cells. Insights into the mechanism of how changes in expression levels of different RNA species affect DSB repair and genome integrity will be conducted in future studies.

  1. The authors do not explain why using this model system is a good idea for the current study. Would these results corroborate in another system?

Response:Thanks for your comments and suggestions. DSBs are the most severe form of DNA damage caused by simultaneous cleavage of double-stranded DNA at opposite or nearby positions by exogenous or endogenous factors. After DSBs occur, chromatin remodeling, cell cycle arrest, apoptosis or other forms of cell death may occur many chemotherapeutic drugs can cause the development of DSB, but BLM is a commonly used inducer of cellular DSBs. The study of other system models may need to be considered separately.

  1. Line 279-280: The authors mention the effect of BLM but they do not corroborate it with previous literature. Dose their effect agree with previous published work?

Response: Thank you for your comments and suggestions. We have updated the reference on line 78-79. Early studies have shown that BLM can affect cell cycle, proliferation, and apoptosis, but the results are inconsistent depending on drug concentration, duration of action, and cell type.

  1. Line 297-298: The authors mention how many GBs of data they have obtained and how much it reduces after “clean-up”. I am not sure that is necessary or relevant as a yardstick of the quality of their data. Rather, it would be useful to know the coverage depth of sequencing and other QC measures.

Response:Thank you for your comments and suggestions. After our consideration, we have made a more detailed description on line 311-315 and 322-325.

  1. In Fig 2c, they mention DDR pathways were identified. This should be highlighted in the plot.

Response:Thank you for your comments and suggestions. We have revised the Figure 2c according to your suggestion.

  1. In section 3.9, the authors just list a number of RNA species identification numbers, but the function/relevance of the particular RNA species does not resonate with the reader. It would be useful to highlight key DDR genes that are known to be regulated by the RNA species identified.

Response:Thank you for your comments and suggestions. Our RNA sequencing results show that differentially expressed mRNAs and non-coding RNAs include known (Supplementary Table S7) and newly discovered ones that have not yet been named. In the process of constructing ceRNA regulatory network, most of them are composed of unnamed RNAs. This new regulatory network can provide a new way to study the mechanism of DSB in bovine cumulus cells in the future.

  1. Since the authors suggest that the altered transcription profiles leads to changes in DDR responses, it would be good to see some experimental validation of some key genes.

Response:Thank you for your comments and suggestions. We will verify key genes and explore their mechanisms in the following studies.

  1. The authors do not provide any mechanistic insights into how the altered transcription levels affect DDR in the cells. It would be imperative to have a paragraph in the discussion explaining possible mechanisms for this phenomenon.

Response: Thank you for your comments and suggestions. After our consideration, we have made a more detailed description on line 559-573 for explaining possible mechanisms of DDR.

Minor comments

  1. The abstract needs to be reworked to bring out the significance of the study

Response:Thank you for your comments and suggestions. We have revised our abstract. We have revised our description on line 21-23.   
2. Font size in all figures to small, labels illegible

Response:Thank you very much for your advice. We have revised our figures according to your suggestion. Due to the limitations of manuscript format, some fonts cannot be adjusted too much. In a normal layout, the picture is visible

  1. Line 21: “these physiological processes”: which process-damage or repair?

Response:Thank you for your comments and suggestions. We have revised our description on line 21.

  1. Line 32: They state that ceRNA help understand the effects of DNA DSB activation. How this is done is unclear. Presumably, the crRNA do not help understand the effect but help in DSB repair?

Response:Thanks for your comments and suggestions. In this study, we established DSB model with BLM induction, and induced changes in cell cycle and proliferation. In previous reports, ceRNA participated in the DSB process, so we sequenced and screened differentially expressed RNAs, and used bioinformatics methods to construct a ceRNA regulatory network to explain the changes in the above physiological phenomena.

  1. In Fig 1b, there are no error bars and statistics on the “G2” bars in the graph

Response: Thank you for your comments and suggestions. In Fig 1b, the standard error of G2 is 0. The following is the original data.

  1. In Fig 1d, there are no statistics shown in the graph

Response:Thank you for your comments and suggestions. Since BLM had no significant effect on the apoptosis rate, no statistical labeling was performed in Figure 1d.

Reviewer 2 Report

The paper aims to analyze and establish a network of ncRNAs when DNA double strand breaks occurs. The manuscript is relevant for the field with relevant references.

However, there are problems that need to be corrected:

- the aim of the paper is insufficiently described and not well detailed

- figures are appropriate for the aim of the paper but they are too small which makes them difficult to understand

- the English language used in the paper is not easy to understand and must undergo revision. Many sentences are unclear and/or incorrect. Here are a few examples. All of them must be corrected and rephrased:

          - lines 21-22 “This study aims to analyze and establish the network of ncRNAs when DSB occurs.”

          - L 37-38 “Mature follicles develop from primordia as well as 37

primary and secondary follicles.” - Primordia is an incorrect name for the follicle development stage.

          - L 55-56 “All of these may cause the quality of oocytes to decline or even die” - It is incorrect to say that the quality of oocytes may die.

          - L 76-78 “. Previous studies have shown …… and oocyte maturation” – this is too long and difficult to understand

- L 109 “CCs were harvested as we described before.” – there is no reference or previous description of this

- the acronym miRNA is not explained in the main text, only in the abstract.

- in section 2.1 no reference is given for harvesting cumulus cells and the “NC group” acronym is not explained in the text

- throughout the paper the number of cumulus cells is written incorrectly

- L 141-142 “removed by centrifugation at 500 x g for 5 min”- description of the protocol is unclear and needs to be clarified

- L 140, L 149 – was anything added to the NC group so that the cells would go through the same treatment as the other group minus the BLM

- section 3.5 - Was BLM added to the cells before assessment of apoptosis?

Author Response

Reviewer #2

Reviewer #2: The paper aims to analyze and establish a network of ncRNAs when DNA double strand breaks occurs. The manuscript is relevant for the field with relevant references.

Response:Thanks for your comments and suggestions. Which are very helpful to improve the quality of this article. We have paid attention to these issues, and the manuscript have been checked thoroughly.

  1. the aim of the paper is insufficiently described and not well detailed.

Response:Thank you for your comments and suggestions. We have revised the abstract to supplement the purpose and significance of the paper.

  1. figures are appropriate for the aim of the paper but they are too small which makes them difficult to understand

Response:Thank you for your comments and suggestions. We have revised our figures to make them clearer. Due to the limitations of manuscript format, some fonts cannot be adjusted too much. In a normal layout, the picture is visible.

  1. the English language used in the paper is not easy to understand and must undergo revision. Many sentences are unclear and/or incorrect. Here are a few examples. All of them must be corrected and rephrased:

- lines 21-22 “This study aims to analyze and establish the network of ncRNAs when DSB occurs.”

- L 37-38 “Mature follicles develop from primordia as well as 37

primary and secondary follicles.” - Primordia is an incorrect name for the follicle development stage.

- L 55-56 “All of these may cause the quality of oocytes to decline or even die” - It is incorrect to say that the quality of oocytes may die.

- L 76-78 “. Previous studies have shown …… and oocyte maturation” – this is too long and difficult to understand

Response:Thank you for your comments and suggestions. We are very sorry for our incorrect writing and we have revised the manuscript.

  1. L 109 “CCs were harvested as we described before.” – there is no reference or previous description of this

Response:Thank you for your comments and suggestions. We have updated the reference 44 on line 111.

  1. the acronym miRNA is not explained in the main text, only in the abstract.

Response: Thank you for your comments and suggestions. We have added the explanation of miRNA to the main text on line 78.

  1. in section 2.1 no reference is given for harvesting cumulus cells and the “NC group” acronym is not explained in the text

Response:Thank you for your comments and suggestions. According to your suggestion, we have supplemented our description about NC group on line 124.

  1. - throughout the paper the number of cumulus cells is written incorrectly

Response:Thank you for your comments and suggestions. We have revised our description.

  1. L 141-142 “removed by centrifugation at 500 x g for 5 min”- description of the protocol is unclear and needs to be clarified

Response:Thanks for your comments and suggestions. We have revised our description on line 143-146.

  1. L 140, L 149 – was anything added to the NC group so that the cells would go through the same treatment as the other group minus the BLM

Response:Thank you for your comments and suggestions. We have revised our description on line 143 and 154. In our study, there were two groups, including cells were treated with 0 μM bleomycin (NC group) or 200 μM bleomycin (BLM group).

  1. section 2.5 - Was BLM added to the cells before assessment of apoptosis?

Response:Thank you for your comments and suggestions. We added the details on line 163 in section 2.5.

Reviewer 3 Report

In the present study, the authors treated bovine cumulus cells with bleomycin (BLM) to construct  double-strand break (DSB) model and performed transcriptome sequencing to find out mRNA, lncRNA, circRNA and miRNA. Finally, two ceRNA networks mediated by lncRNA and circRNA were established. This interesting paper provides a novel insight into the role of DNA DSB activation on the biological function of bovine cumulus cells. The data are of high quality and the paper is clearly written and well reasoned.

Major points:

1. In method section 2.15 CeRNA regulatory network analysis, more details are needed to describe the way to construct the ceRNA networks. That is to say, what are the databases and methods to obtain the ceRNAs? How to predict the binding sites? How to identify the interaction RNAs. The increase of miRNAs should be accompanied by the decrease of both circRNAs, lncRNAs and mRNAs, which should be also recorded.

2. I recommend that some image styles should be uniform to make the results more readable. For example, Figure 2A, 3C, 5A and 6A; Figure 2C, 4B and 5C.

Author Response

Reviewer #3

Reviewer #3: In the present study, the authors treated bovine cumulus cells with bleomycin (BLM) to construct  double-strand break (DSB) model and performed transcriptome sequencing to find out mRNA, lncRNA, circRNA and miRNA. Finally, two ceRNA networks mediated by lncRNA and circRNA were established. This interesting paper provides a novel insight into the role of DNA DSB activation on the biological function of bovine cumulus cells. The data are of high quality and the paper is clearly written and well reasoned..

Response:Thanks for your comments and suggestions, we have revised our manuscript, and we hope that this time the revision can be accepted for publication in Genes.

  1. In method section 2.15 CeRNA regulatory network analysis, more details are needed to describe the way to construct the ceRNA networks. That is to say, what are the databases and methods to obtain the ceRNAs? How to predict the binding sites? How to identify the interaction RNAs. The increase of miRNAs should be accompanied by the decrease of both circRNAs, lncRNAs and mRNAs, which should be also recorded.

Response:Thank you for your comments and suggestions. We added the details on line 264-270 in section 2.15.

  1. I recommend that some image styles should be uniform to make the results more readable. For example, Figure 2A, 3C, 5A and 6A; Figure 2C, 4B and 5C.

Response:Thank you for your comments and suggestions. We have modified and unified the image style of the figures.

Round 2

Reviewer 1 Report

The authors have addressed the concerns adequately.

Reviewer 2 Report

The paper can be published after checking the spelling.